# HLA-B*27:05 alters immunodominance hierarchy of universal influenza-specific CD8+ T cells

Sneha Sant[1], Sergio M. Quiñones-Parra[1¤a], Marios Koutsakos[1], Emma J. Grant[1¤b], Thomas Loudovaris[2], Stuart I. Mannering[2], Jane Crowe[3], Carolien E. van de Sandt[1,4], Guus F. Rimmelzwaan[5¤c], Jamie Rossjohn[6,7,8], Stephanie Gras[6,7], Liyen Loh[1¤d], Thi H. O. Nguyen[1☯*], Katherine Kedzierska[1☯*]

1 Department of Microbiology and Immunology, University of Melbourne, at the Peter Doherty Institute for Infection and Immunity, Parkville, Victoria, Australia, 2 Immunology and Diabetes Unit, St Vincent's Institute of Medical Research, Fitzroy, Victoria, Australia, 3 Deepdene Surgery, Deepdene, Victoria, Australia, 4 Department of Hematopoiesis, Sanquin Research and Landsteiner Laboratory, Amsterdam UMC, University of Amsterdam, Amsterdam, Netherlands, 5 National Influenza Center and Department of Viroscience, Erasmus Medical Center, Rotterdam, The Netherlands, 6 Infection and Immunity Program & Department of Biochemistry and Molecular Biology, Biomedicine Discovery Institute, Monash University, Clayton, Victoria, Australia, 7 Australian Research Council Centre of Excellence for Advanced Molecular Imaging, Monash University, Clayton, Victoria, Australia, 8 Institute of Infection and Immunity, Cardiff University School of Medicine, Heath Park, Cardiff, United Kingdom

☯ These authors contributed equally to this work.
¤a Current address: Department of Molecular Biology, University of California, San Diego, California, United States of America
¤b Current address: Infection and Immunity Program & Department of Biochemistry and Molecular Biology, Biomedicine Discovery Institute, Monash University, Clayton, Victoria, Australia.
¤c Current address: Research Center for Emerging Infections and Zoonoses, University of Veterinary Medicine, Hannover, Germany.
¤d Current address: Department of Immunology and Microbiology, School of Medicine, University of Colorado, Denver, Colorado, United States of America.
* thonguyen@unimelb.edu.au (THON); kkedz@unimelb.edu.au (KK)

**Data Availability Statement:** Data are within the manuscript and its Supporting Information files.

**Funding:** This work was supported by an NHMRC Program Grant (1071916) to KK. KK is an NHMRC

## Abstract

Seasonal influenza virus infections cause 290,000–650,000 deaths annually and severe morbidity in 3–5 million people. CD8+ T-cell responses towards virus-derived peptide/ human leukocyte antigen (HLA) complexes provide the broadest cross-reactive immunity against human influenza viruses. Several universally-conserved CD8+ T-cell specificities that elicit prominent responses against human influenza A viruses (IAVs) have been identified. These include HLA-A*02:01-M1$_{58-66}$ (A2/M1$_{58}$), HLA-A*03:01-NP$_{265-273}$, HLA-B*08:01-NP$_{225-233}$, HLA-B*18:01-NP$_{219-226}$, HLA-B*27:05-NP$_{383-391}$ and HLA-B*57:01-NP$_{199-207}$. The immunodominance hierarchies across these universal CD8+ T-cell epitopes were however unknown. Here, we probed immunodominance status of influenza-specific universal CD8+ T-cells in HLA-I heterozygote individuals expressing two or more universal HLAs for IAV. We found that while CD8+ T-cell responses directed towards A2/M1$_{58}$ were generally immunodominant, A2/M1$_{58}$+CD8+ T-cells were markedly diminished (subdominant) in HLA-A*02:01/B*27:05-expressing donors following *ex vivo* and *in vitro* analyses. A2/M1$_{58}$+CD8+ T-cells in non-HLA-B*27:05 individuals were immunodominant, contained optimal public TRBV19/TRAV27 TCRαβ clonotypes and displayed highly polyfunctional and

Senior Research Level B Fellow. SS was supported by the Victoria-India Doctoral Scholarship (VIDS) and Melbourne International Fee Remission Scholarship (MIFRS). EJG Is supported by an NHMRC CJ Martin Fellowship. C.E.S. has received funding from the European Union's Horizon 2020 research and innovation program under the Marie Skłodowska-Curie grant agreement No. 792532 and University of Melbourne McKenzie Fellowship laboratory support. SG is supported by an NHMRC Senior Research Fellowship. JR was supported by an ARC Laureate fellowship. The funders had no role in study design, data collection and analysis, decision to publish, or preparation of the manuscript.

**Competing interests:** The authors have declared that no competing interests exist.

proliferative capacity, while A2/M1$_{58}$+CD8+ T cells in HLA-B*27:05-expressing donors were subdominant, with largely distinct TCRαβ clonotypes and consequently markedly reduced avidity, proliferative and polyfunctional efficacy. Our data illustrate altered immunodominance patterns and immunodomination within human influenza-specific CD8+ T-cells. Accordingly, our work highlights the importance of understanding immunodominance hierarchies within individual donors across a spectrum of prominent virus-specific CD8+ T-cell specificities prior to designing T cell-directed vaccines and immunotherapies, for influenza and other infectious diseases.

## Author summary

Annual influenza infections cause significant morbidity and morbidity globally. Established T-cell immunity directed at conserved viral regions provides some protection against influenza viruses and promotes rapid recovery, leading to better clinical outcomes. Killer CD8+ T-cells recognising viral peptides in a context of HLA-I glycoproteins, provide the broadest ever reported immunity across distinct influenza strains and subtypes. We asked whether the expression of certain HLA-I alleles affects CD8+ T cells responses. Our study clearly illustrates altered immunodominance hierarchies and immunodomination within broadly-cross-reactive influenza-specific CD8+ T-cells in individuals expressing two or more universal HLA-I alleles, key for T cell-directed vaccines and immunotherapies.

## Introduction

Seasonal influenza virus infections cause 290,000–650,000 deaths annually and severe morbidity in 3–5 million people [1]. Currently licensed vaccines induce strain-specific antibodies but fail to induce influenza-specific CD8+ T cell responses [2]. Furthermore, current vaccines provide little or no protection in the face of a pandemic, caused by the emergence of new influenza A virus (IAV) subtypes, as observed in the most recent 2009 pandemic-H1N1 outbreak [3]. Therefore, in the absence of cross-protective neutralizing antibodies, an efficient way to counteract a novel influenza strain is by re-calling pre-existing, cross-strain-protective cytotoxic CD8+ T cells [4–6].

Protection from influenza-specific memory CD8+ T cells is directed towards more conserved internal viral proteins, such as matrix protein 1 (M1) and nucleoprotein [7–9]. Thus, memory CD8+ T cell responses generated by seasonal IAV infection can provide broader cross-protection against subsequent challenges from distinct influenza virus strains and subtypes, also called heterosubtypic immunity [4, 5, 10, 11]. In humans, CD8+ T cell cross-reactivity between pandemic H1N1-2009 and H3N2 [12], and between the two pandemics H1N1-2009 and H1N1-1918 [10], led to a robust re-call of pre-existing CD8+ T cell immunity towards the newly emerging avian H7N9-2013 strain [5, 13], providing evidence for pre-existing heterosubtypic immunity [14]. Our recent studies also revealed that influenza-specific CD8+ T cells can provide unprecedented immunity across all influenza A, B and C viruses capable of infecting humans [15], and similarly across influenza A [16, 17] and influenza B viruses [15]. These studies demonstrate that pre-existing CD8+ T cell immunity could reduce disease severity, decrease viral burden, ameliorate morbidity and mortality, leading to a rapid recovery of the host. Thus, cross-strain protective CD8+ T cell-based vaccines could provide

life-saving therapeutic strategies for novel reassorting influenza strains with pandemic potential.

During viral infection, CD8$^+$ T cells recognize viral peptides, typically 8–10 amino acids long, that are presented on HLA class I (HLA-I) molecules on the surface of virally-infected cells [18]. Some peptides are highly antigenic and stimulate high magnitude CD8$^+$ T cell responses, termed "immunodominant", while others elicit "subdominant" responses. Thus, overall CD8$^+$ T cell responses directed against multiple immunogenic peptides give rise to immunodominance hierarchy patterns. Immunodominance hierarchies can be affected by several factors, including naïve precursor frequencies, CD8$^+$ T cell receptor (TCR) repertoires capable of generating primary and memory CD8$^+$ T cells, killing capacity, effector polyfunctionality, and TCR avidity for peptide-HLA complexes [19–21]. These factors are further complicated by HLA polymorphisms observed in the human population [20].

CD8$^+$ T cell immunodominance hierarchies in humans has been previously characterized in HIV [22] and CMV [23]. However, immunodominance hierarchies in the context of IAV infection across different HLAs are less clear. In 104 HIV-1-infected patients, CD8$^+$ T cell responses towards otherwise known immunodominant HIV-I-derived peptides presented on HLA-A1, -A2, -A3 and -A24, were reduced in the presence of HLA-B27 and HLA-B57 CD8$^+$ T cell responses, indicating immunodomination of HLA-B27/B57 over other HLA types during HIV-1 infection [22]. Their protective role in delaying disease progression towards AIDS during HIV-1 infection has also been documented [24–26]. To understand factors governing immunodominance patterns in IAV, immunodominance hierarchies need to be defined for known immunodominant IAV epitopes, followed by investigation of the determinants of CD8$^+$ T cell immunodominance across different HLAs.

HLA-A$^*$02:01 is the most prevalent allele found across multiple ethnicities worldwide, including Caucasians (25%), Mexican Seri (54%) and Native North Americans (22%) [5, 17, 27]. Considered to be the most immunodominant IAV epitope is the HLA-A$^*$02:01-restricted $M1_{58-66}$ epitope (hereafter $A2/M1_{58}$), which is universally conserved within influenza viruses circulating over the last century [17]. We have previously defined 5 other universal IAV epitopes presented by common HLA class I types (HLA-A$^*$03:01-$NP_{265-273}$, -B$^*$27:05-$NP_{383-391}$, -B$^*$57:01-$NP_{199-207}$, -B$^*$18:01-$NP_{219-226}$ and -B$^*$08:01-$NP_{225-233}$) [5], of which to date, there have been no extensive studies examining their CD8$^+$ T cell immunodominance hierarchies.

Here, we investigated immunodominant $A2/M1_{58}$ responses in heterozygote individuals expressing the other universal HLAs for IAVs. We analyzed T cell polyfunctionality, proliferation kinetics, peptide/HLA-I avidity and direct $ex$ $vivo$ TCRαβ repertoire to determine how these factors are impacted by HLA-A$^*$02:01 responses in the presence of different HLA-I molecules.

## Results

### $A2/M1_{58}^+CD8^+$ T cell immunodominance hierarchy is altered by HLA-B$^*$27:05 expression

To define the immunodominance hierarchy across universal influenza CD8$^+$ T cell epitopes (HLA-A$^*$02:01-$M1_{58-66}$, HLA-A$^*$03:01-$NP_{265-273}$, HLA-B$^*$27:05-$NP_{383-391}$, HLA-B$^*$57:01-$NP_{199-207}$, HLA-B$^*$18:01-$NP_{219-226}$ and HLA-B$^*$08:01-$NP_{225-233}$) [5] within an individual, IAV-specific CD8$^+$ T cell responses towards those epitopes were measured in healthy blood donors following peptide stimulation and IFN-γ/TNF cytokine production (Fig 1A). Robust IFN-γ$^+$CD8$^+$ T cell responses were readily detected across all 6 conserved epitopes: $A2/M1_{58}$ (mean = 9.4%, range = 0.1–31.6%), $A3/NP_{265}$ (8.3%, 0.1–24.1%), $B8/NP_{225}$ (3.0%, 0.1–7.2%), $B18/NP_{219}$ (13.1%, 7.0–15.9%), $B27/NP_{383}$ (16.7%, 4.5–41.1%) and $B57/NP_{199}$ (9.98%, 7.3%-14.6%), confirming the immunogenicity of these highly conserved epitopes (Fig 1B).

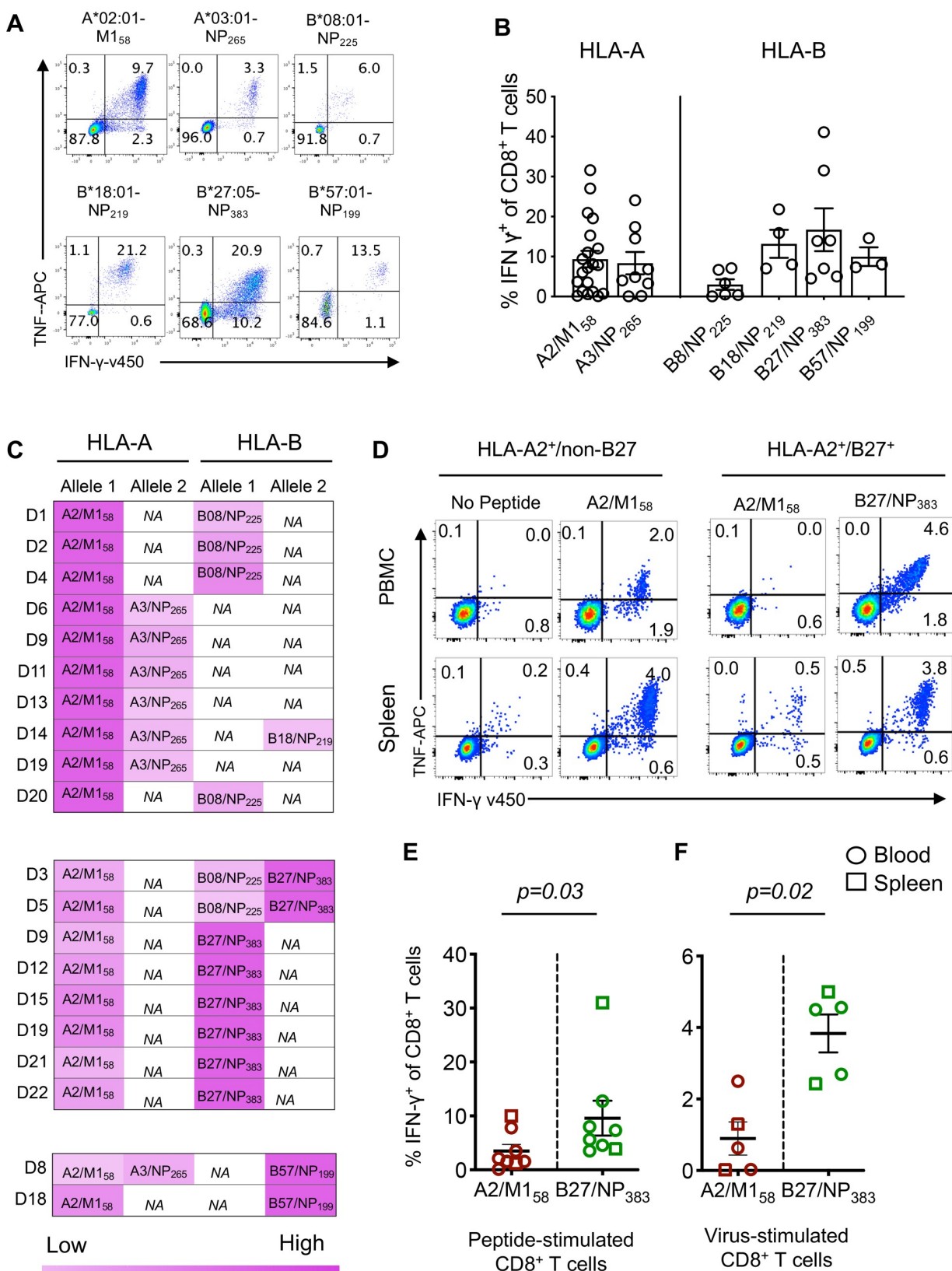

**Fig 1. Reduced magnitude of A2/M158+CD8+ T cell responses in HLA-A2+/B27+ individuals.** (**A**) Representative FACS plots of cytokine production following IFN-γ/TNF ICS assay of day 10 peptide-expanded T cell lines from healthy PBMCs for each universal epitope. (**B**) Frequency of IFN-γ-producing CD8+ T cells from T cell lines expanded: A2/M1$_{58}$ (n = 20), A3/NP$_{265}$ (n = 9), B8/NP$_{225}$ (n = 6), B18/NP$_{219}$ (n = 4), B27/NP$_{383}$ (n = 7) and B57/NP$_{199}$ (n = 3). Bars represent mean±SEM. (**C**) Heatmap of relative contribution of IFN-γ+CD8+ T cell responses across different epitopes restricted by different HLAs in individuals with more than 1 universal HLA allele (n = 20). NA represents the individual who is either homozygote or has a non-universal HLA at that locus. (**D**) Representative FACS plots comparing A2/M1$_{58}$+CD8+ T cell responses in HLA-A2+/non-B27 individuals versus A2/M1$_{58}$ and B27/NP$_{383}$ responses in HLA-A2+/B27+ individuals. Day 10 A2/M1$_{58}$ and B27/NP$_{383}$ responses in HLA-A2+/B27+ individuals after T cell lines were generated in parallel with (**E**) single peptide-pulsed or (**F**) live PR8 virus-pulsed cells before performing ICS with relevant peptide. Data (mean±SEM) are pooled from blood (open circle) and spleen (open square) donors over 2–3 independent experiments (n = 5–8). Exact *p*-values are shown (Mann-Whitney t test).

The immunodominance patterns in donors co-expressing 2 or more of the universal HLAs were subsequently assessed in human blood and spleen. When we compared the magnitude of IFN-γ+CD8+ T cell responses directed at the most studied immunodominant IAV epitope A2/M1$_{58}$ [5, 17] against other universal epitope responses within the same individual, our analyses revealed that A2/M1$_{58}$+CD8+ T cell responses were immunodominant over HLAs A3, B8 and B18, but this was strikingly not the case in donors that co-expressed B27 or B57 (Fig 1C). As HLA-B*27:05 mediates protection in HIV and HCV [28], and is associated with viral escape mutations in HIV [29] and a slow accumulation of variants within IAV-H3N2 [30], we focused on understanding the immunodominance mechanisms of B27/NP$_{383}$+CD8+ T cell responses over A2/M1$_{58}$+ responses in HLA-A2+/B27+ co-expressed individuals. Although A2/M1$_{58}$+ responses were immunodominant in non-HLA-B27 donors, individuals co-expressing HLA-A2+/B27+ showed significantly higher IFN-γ+CD8+ T cell responses towards B27/NP$_{383}$ (mean = 9.6%, SD±3.2%) in comparison to subdominant A2/M1$_{58}$ responses (3.5%±1.2%) across all the blood and spleen donors tested (Fig 1D and 1E, *p* = 0.03).

To determine whether the B27/NP$_{383}$>A2/M1$_{58}$ immunodominance hierarchy also occurred during influenza virus infection, we measured CD8+ T cell responses in blood and spleen following stimulation with autologous IAV-infected APCs (Fig 1F), rather than peptide-pulsed APCs (Fig 1E), which more closely resembles the natural antigen presentation pathway [20]. Stimulation of PBMCs with virus-infected APCs, followed by a 6-hr ICS verified significantly higher B27/NP$_{383}$+CD8+ T cell responses (3.8%±1.2%) than A2/M1$_{58}$+ responses (0.89%±0.9%) (Fig 1F, *p* = 0.02). Taken together, our data show, that the expression of specific HLAs (like HLA-B*27:05) can lead to immunodomination and markedly reduce immunodominance of universal influenza-specific CD8+ T cells.

## Reduced *ex vivo* frequencies within subdominant A2/M1$_{58}$+CD8+ T cells in A2+B27+ donors

As the experiments presented in Fig 1 were performed using *in vitro* assays, we subsequently determined the frequencies of immunodominant and subdominant influenza-specific CD8+ T cells directly *ex vivo* using a dual tetramer-associated magnetic enrichment (TAME) (Fig 2A), allowing increased detection of epitope-specific T cells by up to 100-fold [31–33]. In accordance with our *in vitro* experiments, subdominant A2/M1$_{58}$+CD8+ T cell frequencies in HLA-A2+/B27+ donors were significantly diminished within both PBMCs (4.9E-05±2.0E-05, *p* = 0.019) and spleen (8.9E-06±3.0E-06, *p* = 0.042), compared to dominant A2/M1$_{58}$+CD8+ T cells (PBMCs: 2.8E-04±9.4E-05; spleen: 4.6E-05 ± 4.7E-06) (Fig 2B). The frequencies of immunodominant B27/NP$_{383}$+CD8+ T cells were trending higher than the subdominant A2/M1$_{58}$+CD8+ T cells in both PBMC and spleen co-expressed donors.

Phenotypic *ex vivo* analyses of immunodominant and subdominant influenza-specific A2/M1$_{58}$+CD8+ T cells involved CCR7 and CD45RA expression to characterize naïve (T$_N$, CCR7+CD45RA+), central memory (T$_{CM}$, CCR7+CD45RA-), effector memory (T$_{EM}$,

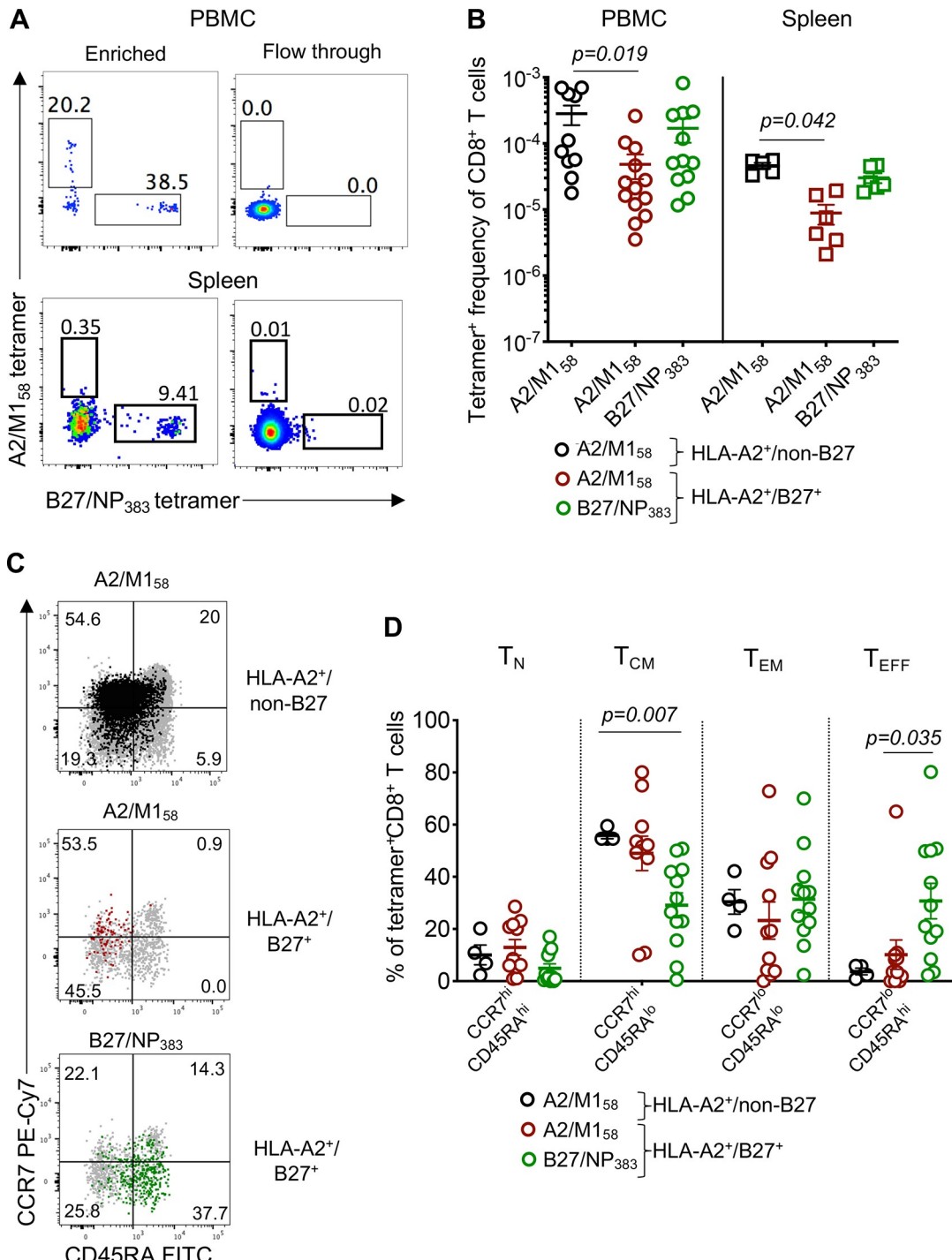

**Fig 2. Ex vivo frequencies and T cell phenotype of A2/M158+ and B27/NP383+ CD8$^+$ T cells.** (**A**) Representative FACS plots of TAME-enriched A2/M1$_{58}$$^+$ and B27/NP$_{383}$$^+$ CD8$^+$ T cells isolated from PBMC and spleen of HLA-A2$^+$/B27$^+$ individuals. Minimal cells were observed in the flow through fractions. Cells were gated on live CD14$^-$CD19$^-$CD3$^+$CD8$^+$ T cells. (**B**) Precursor frequencies of A2/M1$_{58}$$^+$ and B27/NP$_{383}$$^+$ CD8$^+$ T cells from HLA-A2$^+$/B27$^+$ (PBMC n = 13; spleen n = 6) versus A2/M1$_{58}$$^+$CD8$^+$ T cells from HLA-A2$^+$/non-B27 donors (PBMC n = 10; spleen n = 5). (**C**) CCR7 and CD45RA expression profiles of TAME-enriched tetramer$^+$ cells overlaid on top of total CD8$^+$ T cells from each group with at least 10 tetramer$^+$ events. (**D**) Frequencies of naïve (T$_N$, CCR7$^+$CD45RA$^+$), central memory (T$_{CM}$, CCR7$^+$CD45RA$^-$), effector memory (T$_{EM}$, CCR7$^-$CD45RA$^-$) and effector (T$_{EFF}$, CCR7$^-$CD45RA$^+$) T cell populations for immunodominant A2/M1$_{58}$$^+$ (black, n = 4), subdominant A2/M1$_{58}$$^+$ (maroon, n = 11) and immunodominant B27/NP$_{383}$$^+$ (green, n = 12) specificities. Data are pooled from blood and spleen donors over 4–5 independent experiments. (**B** and **D**) Bars show mean±SEM and statistically significant exact $p$-values are shown ($p<0.05$, Kruskal-Wallis test, one-way ANOVA).

CCR7⁻CD45RA⁻) and effector (T$_{EFF}$, CCR7⁻CD45RA⁺) T cells (Fig 2C). The memory populations were relatively similar between immunodominant and subdominant A2/M1$_{58}$⁺CD8⁺ T cells. However, immunodominant B27/NP$_{383}$⁺CD8⁺ T cells were significantly enriched for effector cells and with diminished central memory phenotype in comparison to both immunodominant and subdominant A2/M1$_{58}$⁺CD8⁺ T cells, which were predominantly of central memory rather than effector phenotype (Fig 2D), suggesting different transitioning between B27/NP$_{383}$⁺CD8⁺ and A2/M1$_{58}$⁺CD8⁺ T cell memory subsets.

## Distinct A2/M1$_{58}$⁺CD8⁺ TCRαβ repertoires in the presence of HLA-B*27:05

Altered *ex vivo* frequencies and phenotypes of A2/M1$_{58}$⁺CD8⁺ T cells in A2/B27-expressing donors suggested possible underlying differences in TCRαβ repertoire composition and diversity between immunodominant and subdominant A2/M1$_{58}$⁺CD8⁺ T cells. To dissect TCRαβ repertoires, we single-cell isolated TAME-enriched subdominant A2/M1$_{58}$⁺CD8⁺ T cells directly *ex vivo* from three HLA-A2⁺/B27⁺ donors for TCR analyses and compared them to our previously defined immunodominant A2/M1$_{58}$⁺ TCRαβ repertoires from HLA-A2⁺/non-HLA-B27 donors [17], which predominantly consisted of the TRBV19 and TRAV27 gene segments and the prominent public TRBV19/TRAV27 TCRαβ clonotype: CDR3α-GGSQGNL and CDR3β-SSIRSYEQ [17].

Notably, dissection of TRBV and TRAV gene usage revealed that subdominant A2/M1$_{58}$⁺CD8⁺ T cells did not predominantly consist of the public TRBV19 and TRAV27 gene segments but encompassed a diverse array of TRBV and TRAV genes, with one donor (SD1) displaying neither TRBV19 nor TRAV27 (Fig 3A, Table 1). In fact, the public A2/M1$_{58}$⁺ TRBV19/TRAV27 clonotype was only present in two HLA-A2⁺/B27⁺ donors at low frequencies (SD2 20% and SD3 10.2%) (Fig 3B and 3C), compared to high public TCRαβ clonotype frequencies observed in all HLA-A2⁺/non-HLA-B27 donors (average 48%, range 15–67%), as previously described [17]. Moreover, TRBV19 and, more significantly TRAV27 gene usage, were lower in the subdominant A2/M1$_{58}$⁺ TCRαβ repertoires compared to the known immunodominant repertoires (TRAV27: $p = 0.02$) (Fig 3B).

Further analysis of a CDR3 length usage (Fig 3D) demonstrated a minor difference in the preferred CDR3β-chain length (8 aa) between immunodominant (87%±9%) [17] and subdominant A2/M1$_{58}$⁺CD8⁺ T cells (51%±20.6%, SD2 preferred 9 aa length). Strikingly, subdominant A2/M1$_{58}$⁺CD8⁺ T cells had a preference for a longer CDR3α length of 9 aa (41.6%±13.2%), as compared to a shorter 7 aa length preference in immunodominant donors (48.8%±34.7%) [17].

Structural studies have defined the CDR3β motif "RS" as an essential feature for the "peg-notch" recognition by the public TRBV19/TRAV27 TCR of the "plain vanilla" A2/M1$_{58}$ epitope [34]. Subdominant A2/M1$_{58}$⁺CD8⁺ T cells had a considerable reduction in the CDR3β "RS" motif (Table 1, 56.5%±2.19%, 18/43 unique TCRs across 2 donors, 1 donor had no "RS" motif) in comparison to immunodominant A2/M1$_{58}$⁺CD8⁺ T cells (82.3%±15.9%, 20/33 unique TCRs across 3 donors) [17]. Furthermore, the Simpson's Diversity Index (values 0 to 1, with 1 being the most diverse) for subdominant A2/M1$_{58}$⁺CD8⁺ TCRs (0.9±0.1) was on average ~25% more diverse that immunodominant A2/M1$_{58}$⁺CD8⁺ TCRs (0.7±0.3).

Overall, the influenza-specific A2/M1$_{58}$⁺CD8⁺ TCRαβ repertoire revealed striking differences in TCRαβ clonal composition and diversity between subdominant (in HLA-B*27:05-expressing donors) and immunodominant (in non-HLA-B*27:05 individuals) A2/M1$_{58}$⁺CD8⁺ T cells, including reduced TRAV27 and TRBV19 "RS" motif usage, lower occurrence of the public TRBV19/TRAV27 clonotype, increased diversity and increased prevalence of private

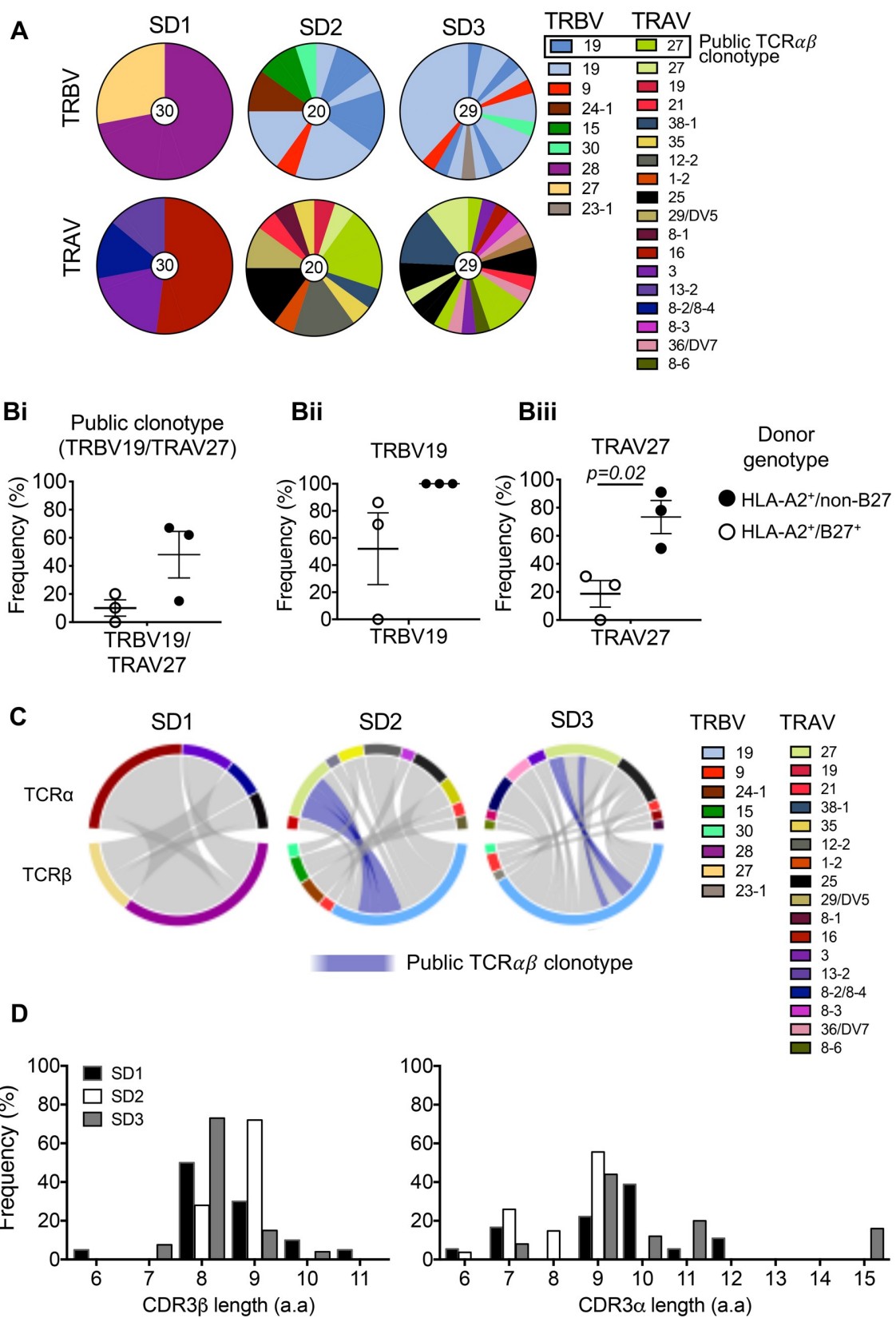

**Fig 3. Subdominant A2/M158+ TCRαβ repertoire display reduction in A2/M1$_{58}$-specific "public motif".** (**A**) Pie charts of TRAV and TRBV usage within subdominant A2/M1$_{58}^+$CD8$^+$ T cells in three HLA-A2$^+$/B27$^+$ blood donors. Public TRBV19 and TRAV27 motifs are represented by blue and green shades respectively. (**B**) Frequencies of the dominant A2/M1$_{58}$-specific gene usage are shown for paired TRBV19/TRAV27 or TRBV19 and TRAV27 groups for our three subdominant HLA-A2$^+$/B27$^+$ donors (SD1-3, open circles) versus previously reported immunodominant HLA-A2$^+$/non-B27 donors (closed circles) [17]. Mean±SEM are shown. (**C**) Circos plots made in RStudio v.1.2.1335 of paired TCRαβ clonotypes from SD1-3 donors. (**D**) Frequencies of CDR3β and CDR3α aa lengths are shown for 3 our subdominant donors SD1-3.

clonotypes. These differences in TCRαβ repertoires between A2/B27- and A2/non-B27-expressing donors can, at least in part, explain differential immunodominance patterns of influenza-responding CD8$^+$ T cells directed at the A2/M1$_{58}$ epitope.

## Diminished polyfunctional capacity of subdominant A2/M1$_{58}^+$ CD8$^+$ T cells

Having found altered immunodominance hierarchy of A2/M1$_{58}^+$CD8$^+$ T cells in the presence of HLA-B$^*$27:05 expression, associated with strikingly different TCRαβ repertoires, we subsequently assessed the consequence of such changes on the quality and functionality of A2/M1$_{58}^+$CD8$^+$ T cell responses in both HLA-A2$^+$/non-B27 and HLA-A2$^+$/B27$^+$ individuals. We defined their ability to simultaneously produce multiple cytokines, defined as polyfunctionality [35], such as IFN-γ, TNF and the degranulation/cytotoxicity marker CD107a (Fig 4A). A2/M1$_{58}^+$CD8$^+$ T cells displayed significantly reduced functional profiles (~45% >1 functional response) when compared to B27/NP$_{383}^+$CD8$^+$ T cells (~78%) in HLA-A2$^+$/B27$^+$ individuals (Fig 4B, $p$ = 0.0002), which also had strikingly higher frequencies of triple cytokine producers. (Fig 4C, $p$ = 0.003). Similarly, these subdominant A2/M1$_{58}^+$CD8$^+$ T cells had significantly reduced polyfunctionality compared to the those in HLA-A2$^+$/non-B27 donors (~64%) (Fig 4B, $p$ = 0.007), and a trend towards a reduction in triple cytokine producers (Fig 4C). Interestingly, the total CD107a$^+$A2/M1$_{58}^+$ response remained unchanged between the groups (Fig 4B, magenta arc), suggesting that A2/M1$_{58}^+$CD8$^+$ T cells maintain degranulation capacity independent of HLA-B27 co-expression. Expression of effector molecules, granzyme B and perforin, were also measured, both ~80% expressed.

## Subdominant A2/M1$_{58}^+$ CD8$^+$ T cells have lower functional T cell avidity

Antigen sensitivity is a major readout for CD8$^+$ T cell functional avidity [36]. To assess the functional avidity between immunodominant and subdominant influenza-specific CD8$^+$ T cells, the level of antigen sensitivity was determined by re-stimulating T cell lines with 10-fold dilutions of peptide and measuring the EC$_{50}$ of the total IFN-γ response (Fig 5A). Our data showed that immunodominant B27/NP$_{383}^+$ and A2/M1$_{58}^+$ CD8$^+$ T cells (mean EC$_{50}$: $10^{-10}$ M and $10^{-10}$ M, respectively) were 10 times more sensitive to the peptide antigen stimulation compared to subdominant A2/M1$_{58}^+$ CD8$^+$ T cells (mean EC50: $10^{-9}$ M) (Fig 5B). However, the functional avidity EC$_{50}$ values of subdominant A2/M1$_{58}^+$CD8$^+$ T cells are just above the optimal nanomolar to picomolar physiological ranges [37], suggesting that they could still mount an immune response, albeit at lower magnitudes. The differences in antigen sensitivity between immunodominant and subdominant CD8$^+$ T cells of the same specificity can be explained, at least partially, by different TCRαβ clonotypes within these immunodominant and subdominant A2/M1$_{58}^+$CD8$^+$ T cells, highlighting the basis for distinct patterns of immunodominance hierarchy.

## Reduced proliferative capacity within subdominant A2/M1$_{58}^+$ versus dominant B27/NP$_{383}^+$ CD8$^+$ T cells

To determine the effect of altered TCRαβ repertoires and pHLA-I avidity on the proliferative capacity of immunodominant B27/NP$_{383}^+$ and subdominant A2/M1$_{58}^+$CD8$^+$ T cells, both A2/

**Table 1. TCRαβ repertoire within subdominant A2/M158+CD8+ T cells.**

| TRBV | TRBJ | CDR3β | CDR3α | TRAV | TRAJ | SD1 | SD2 | SD3 |
|------|------|-------|-------|------|------|-----|-----|-----|
| | | | | | | % | % | % |
| TRBV19 | TRBJ2-1 | SSILAGAYNEQ | ND | TRAV19 | TRAJ22 | | 5 | |
| TRBV19 | TRBJ2-7 | SSIRSAYEQ | GAHGSSNTGKL | TRAV27 | TRAJ37 | | 5 | |
| **TRBV19** | **TRBJ2-3** | **SSIRSSDTQ** | **GAGGGSQGNL** | **TRAV27** | **TRAJ42** | | **5** | |
| **TRBV19** | **TRBJ2-7** | **SSIRSSYEQ** | **GGGSGGSQGNL** | **TRAV27** | **TRAJ42** | | **5** | |
| **TRBV19** | **TRBJ2-7** | **SSIRSSYEQ** | **GGGSQGNL** | **TRAV27** | **TRAJ42** | | **10** | |
| TRBV19 | TRBJ2-1 | SSLAGPYNEQ | FMTATFTSGTYKY | TRAV38-1 | TRAJ40 | | 5 | |
| TRBV19 | TRBJ1-1 | SSPQGGAEA | GQIXXRAGXXL | TRAV35 | TRAJ5 | | 5 | |
| TRBV19 | TRBJ2-1 | SSPRSALEQ | VNWGGGSQGNL | TRAV12-2 | TRAJ42 | | 15 | |
| TRBV9 | TRBJ1-5 | SSPWDRGQPQ | VEXXXXKI | TRAV1-2 | TRAJ30 | | 5 | |
| TRBV19 | TRBJ1-4 | SSVRSDEKL | GNYGGSQGNL | TRAV25 | TRAJ42 | | 15 | |
| TRBV24-1 | TRBJ1-6 | TSIAPI | APPNSGNTPL | TRAV29/DV5 | TRAJ29 | | 10 | |
| TRBV15 | TRBJ2-7 | TSKSGGPYEQ | PKGYSTL | TRAV21 | TRAJ11 | | 5 | |
| TRBV15 | TRBJ1-1 | TSRDLGVWTEA | VKGPYGGGSQGNL | TRAV8-1 | TRAJ42 | | 5 | |
| TRBV30 | TRBJ2-3 | ND | GIPSTGANSKL | TRAV35 | TRAJ56 | | 5 | |
| TRBV28 | TRBJ2-7 | SSVFGTSYEQ | LPSCSGNTPL | TRAV16 | TRAJ29 | 45 | | |
| TRBV28 | TRBJ2-7 | SSVFGTSYEQ | LPSCSGNTPL | TRAV16 | TRAJ15 | 7 | | |
| TRBV28 | TRBJ2-7 | SSVFGTSYEQ | VSNQAGTAL | TRAV3 | TRAJ15 | 17 | | |
| TRBV28 | TRBJ2-7 | SSVFGTSYEQ | C##NQAGTAL | TRAV3 | TRAJ15 | 3 | | |
| TRBV27 | TRBJ1-1 | SSYGQGLEA | APNDYKL | TRAV8-2/TRAV8-4 | TRAJ37 | 14 | | |
| TRBV27 | TRBJ1-1 | SSYGQGLEA | AXAT#GKL | TRAV13-2 | TRAJ37 | 3 | | |
| TRBV27 | TRBJ1-1 | SSYGQGLEA | AXATXAX# | TRAV13-2 | TRAJ37 | 11 | | |
| **TRBV19** | **TRBJ2-7** | **SIRSSYEQ** | **GSGGSQGNL** | **TRAV27** | **TRAJ42** | | | **3.4** |
| TRBV19 | TRBJ2-3 | GTGSIDTQ | RDGTGANNL | TRAV3 | TRAJ36 | | | 3.4 |
| TRBV19 | TRBJ1-2 | SFGSYGY | RATSGGSNYKL | TRAV16 | TRAJ53 | | | 3.4 |
| TRBV19 | TRBJ2-7 | SIRSSYEQ | ND | TRAV8-3 | TRAJ44 | | | 3.4 |
| TRBV19 | TRBJ2-2 | SARSTGEL | EPKG#TGANNL | TRAV36/DV7 | TRAJ36 | | | 3.4 |
| TRBV9 | TRBJ1-5 | SVEGNQPQ | REYMGSSYKL | TRAV14/DV4 | TRAJ12 | | | 3.4 |
| TRBV19 | TRBJ2-2 | SARSTGEL | NYGGSQGNL | TRAV25 | TRAJ42 | | | 6.9 |
| TRBV30 | TRBJ2-3 | SVAGGPGDTQ | VPMEYGNKL | TRAV21 | TRAJ47 | | | 3.4 |
| TRBV19 | TRBJ2-2 | SARSTGEL | ND | TRAV36/DV7 | TRAJ42 | | | 3.4 |
| TRBV19 | TRBJ1-5 | SLFSQQPQ | VYGGSQGNL | TRAV27 | TRAJ42 | | | 3.4 |
| TRBV19 | TRBJ2-2 | SVRSTGEL | GSGGSQGNL | TRAV27 | TRAJ42 | | | 3.4 |
| **TRBV19** | **TRBJ1-1** | **SIRSSYEA** | **NYGGSQGNL** | **TRAV27** | **TRAJ42** | | | **3.4** |
| TRBV19 | TRBJ2-1 | STRSGDEQ | WNQGGKL | TRAV8-6 | TRAJ23 | | | 3.4 |
| TRBV23-1 | TRBJ2-2 | NA | RDGTGANNL | TRAV3 | TRAJ36 | | | 3.4 |
| TRBV19 | TRBJ2-2 | SARSTGEL | ND | TRAV36/DV7 | TRAJ42 | | | 3.4 |
| **TRBV19** | **TRBJ2-7** | **SIRSSYEQ** | **GGSQGNL** | **TRAV27** | **TRAJ42** | | | **3.4** |
| TRBV9 | TRBJ2-2 | ND | NYGGSQGNL | TRAV25 | TRAJ42 | | | 3.4 |
| TRBV19 | TRBJ2-2 | STRSTGEL | NYGGSQGNL | TRAV25 | TRAJ42 | | | 3.4 |
| TRBV19 | TRBJ2-4 | DEGSGIQ | ND | TRAV27 | TRAJ31 | | | 3.4 |
| TRBV19 | TRBJ2-3 | SSIRSTDTQ | CNYGGSQGNL | TRAV25 | TRAJ42 | | | 6.9 |
| TRBV19 | TRBJ1-2 | SSTGSYGY | AFMINAGGTSYGKLT | TRAV38-1 | TRAJ52 | | | 13.8 |
| TRBV19 | TRBJ2-7 | SSVRSAYEQ | GAIGSSNTGKL | TRAV27 | TRAJ37 | | | 10.3 |
| **No. of sequences** | | | | | | **30** | **20** | **29** |

Bold = public TCR, underlined = public "RS" CDR3β motif, ND = not determined, X = undefined aa sequence, # = out-of-frame sequence resulting in unproductive pair. SD = donor code.

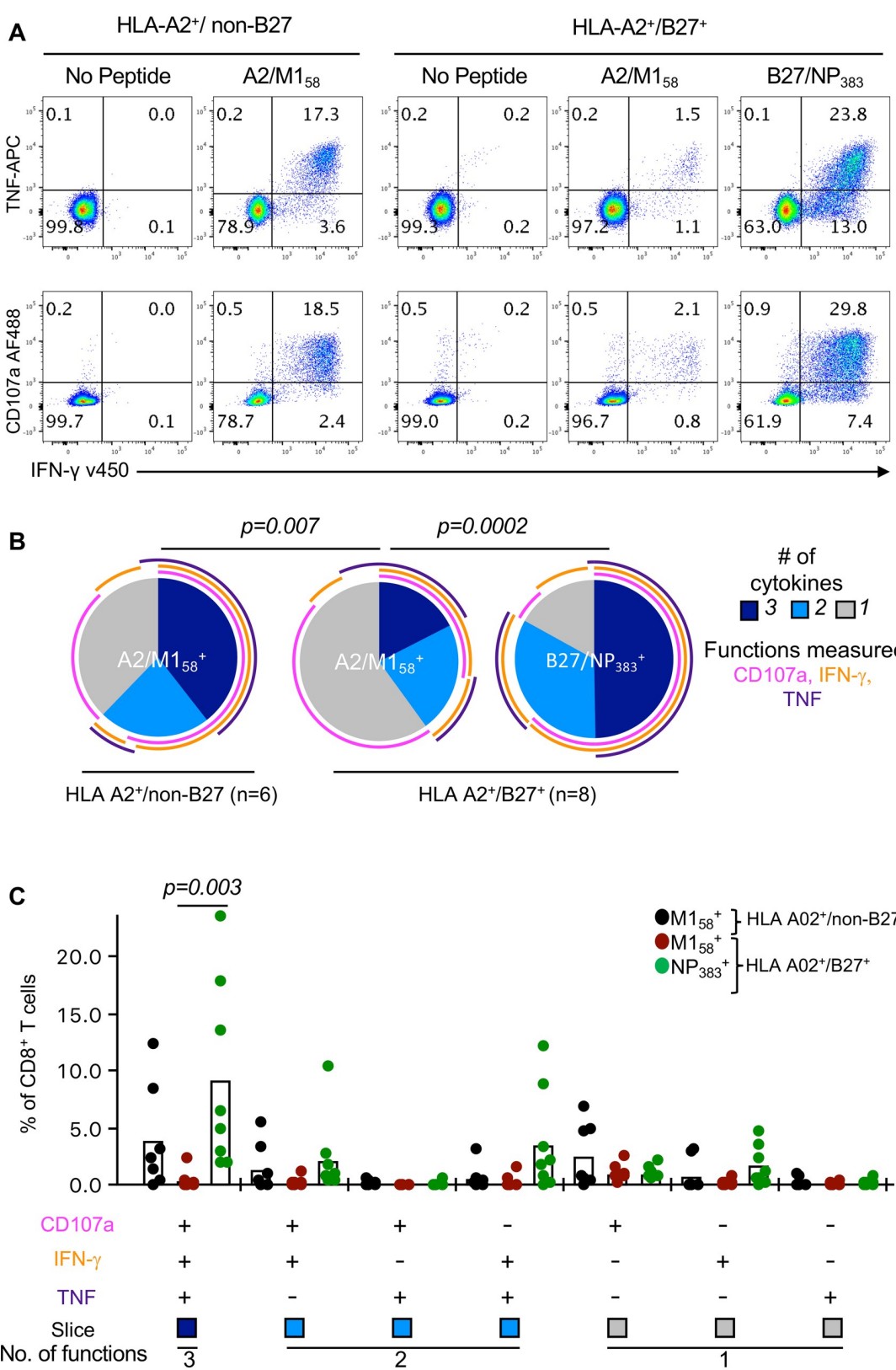

**Fig 4. Diminished polyfunctionality of A2/M158+CD8$^+$ T cells in HLA-A2$^+$/B27$^+$ individuals.** (**A**) Representative FACS plots of CD107a, IFN-γ and TNF responses of day 10 peptide-expanded CD8$^+$ T cell lines from HLA-A2$^+$/non-B27 and HLA-A2$^+$/B27$^+$ donors. (**B**) Pie charts representing the average fractions of expanded CD8$^+$ T cells expressing different combinations of CD107a, IFN-γ and TNF were generated using Pestle v1.8 and SPICE v5.35 software. Arcs represent average frequency of individual cytokines. Exact *p*-values are calculated using Spice permutation test (10,000) replicates. (**C**) Frequencies of the CD8$^+$ T cell functional response producing different combinations of CD107a, IFN-γ and TNF. Data (mean ±SEM) are pooled from blood and spleen donors over 2–3 independent experiments (n = 7–8). For each cytokine combination, statistically significant exact *p*-values are shown for each response (*p*<0.05, Kruskal-Wallis test, one-way ANOVA).

M1$_{58}$$^+$CD8$^+$ T cells and B27/NP$_{383}$$^+$CD8$^+$ T cells were stimulated with their cognate peptides, and their functional and proliferative kinetics were assessed via IFN-γ production and peptide/MHC-tetramer staining, respectively (Fig 6A). In all 5 donors tested, the magnitude of proliferating B27/NP$_{383}$$^+$CD8$^+$ T cells was greater than A2/M1$_{58}$$^+$CD8$^+$ T cells, with higher frequencies of IFN-γ producers from day 6 onwards, which was significantly higher on days 9 and 12 (Fig 6B, both *p* = 0.007). To verify the cytokine readout data, enumeration of A2/M1$_{58}$$^+$ and B27/NP$_{383}$$^+$ tetramer$^+$CD8$^+$ T cells showed more rapid expansion in the number of B27/NP$_{383}$-tetramer$^+$CD8$^+$ T cells compared to A2/M1$_{58}$-tetramer$^+$CD8$^+$ T cells from day 9 up to day 12 (average 2.6-fold increase) (Fig 6C). Thus, B27/NP$_{383}$$^+$ CD8$^+$ T cells immunodominance over A2/M1$_{58}$$^+$ CD8$^+$ T cells was associated with higher proliferative capacity of B27/NP$_{383}$$^+$ CD8$^+$ T cells in HLA-A$^*$02:01/HLA-B$^*$27:05-expressing individuals.

Taken together, our data demonstrate that the immunodominant or subdominant status of A2/M1$_{58}$$^+$CD8$^+$ T cells and the underlying differences in TCRαβ repertoires can significantly affect influenza-specific CD8$^+$ T cell polyfunctionality, quality, pMHC avidity and proliferative capacity of A2/M1$_{58}$$^+$ CD8$^+$ T cells. While in HLA-A2$^+$/non-B27 individuals, immunodominant A2/M1$_{58}$$^+$CD8$^+$ T cells are highly abundant, polyfunctional and display high avidity and public TCRαβ repertoires, subdominant A2/M1$_{58}$$^+$CD8$^+$ T cells in HLA-A2$^+$/B27$^+$ co-expressing individuals are of lower frequency, display lower polyfunctional, are less proliferative (compared to B27/NP$_{383}$$^+$ CD8$^+$ T cells), with low avidity TCRαβ repertoires.

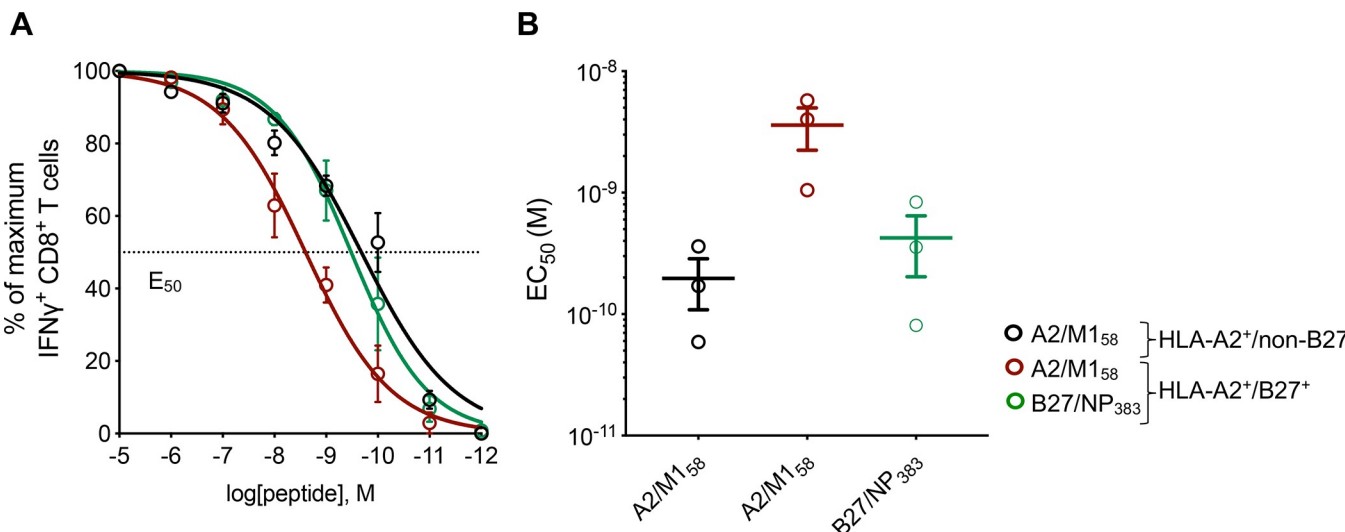

**Fig 5. Reduced antigen sensitivity in subdominant A2/M158+CD8$^+$ T cells.** (**A**) Peptide titration curve and (**B**) EC$_{50}$ values for each T cell specificity after 12 days of peptide expansion. Mean and SEM are shown (n = 3 for each group).

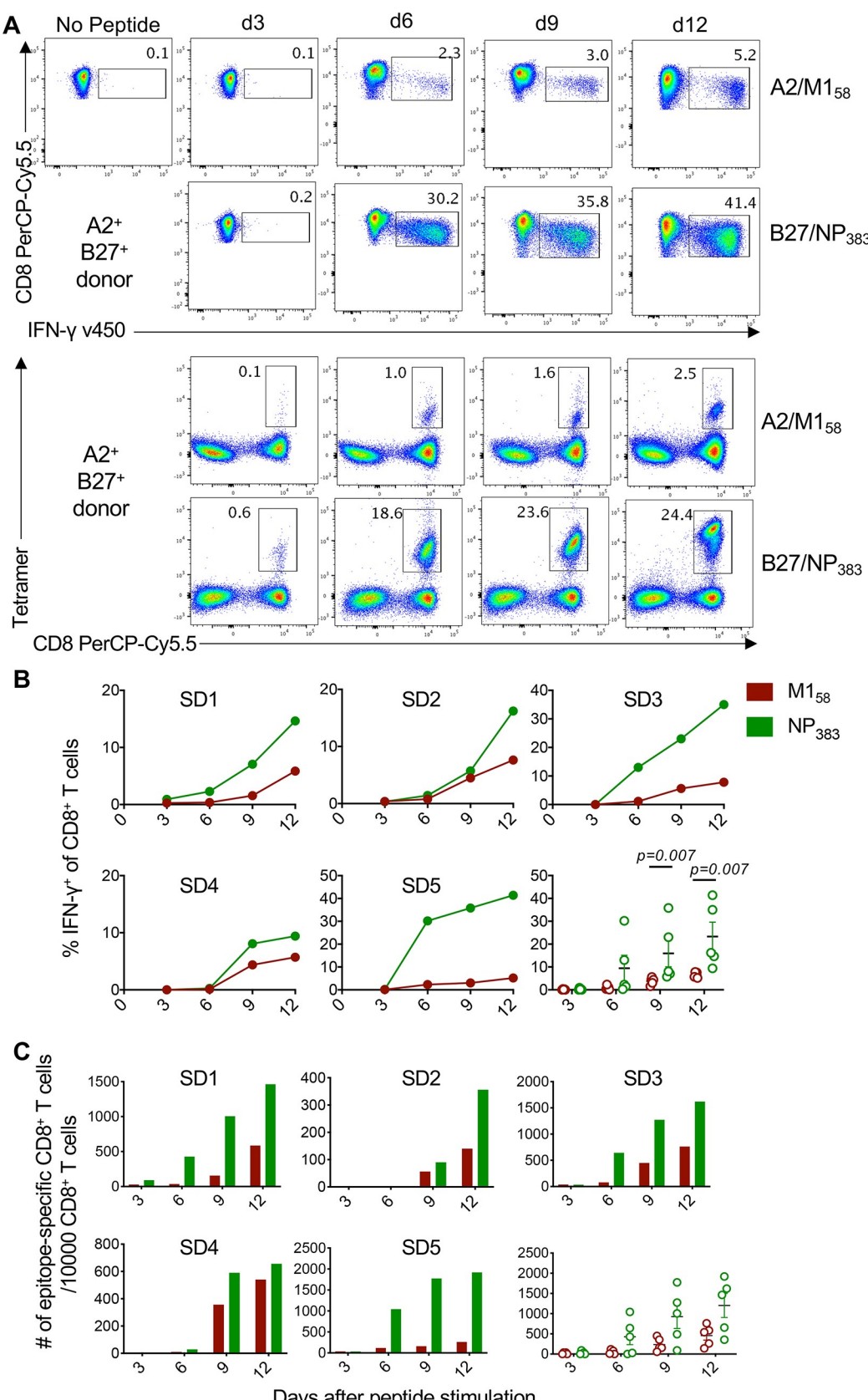

**Fig 6. Higher proliferative capacity of B27/NP383+CD8$^+$ T cells over A2/M158+CD8$^+$ T cells in HLA-A2$^+$/B27$^+$ individuals.** (**A**) Representative kinetics of IFN-$\gamma^+$ and tetramer$^+$ responses of peptide-stimulated T cell lines on days 3, 6, 9 and 12 of culture. (**B**) Parallel IFN-$\gamma^+$ A2/M1$_{58}^+$ and B27/NP$_{383}^+$ responses and (**C**) absolute numbers of tetramer$^+$CD8$^+$ T cells over time for each individual and as a group (mean±SEM). Blood donors: SD1-3; spleen donors: SD4-5. Statistically significant exact *p*-values are shown (Mann Whitney t test).

## Discussion

Our study reports, that immunodominance can occur within human prominent influenza-specific CD8$^+$ T cells with co-expression of another specific HLA class I molecule. While broadly cross-reactive universal A2/M1$_{58}^+$CD8$^+$ T cells are immunodominant, contain the optimal public TRBV19/TRAV27 TCR$\alpha\beta$ repertoire and display highly polyfunctional and proliferative capacity in non-HLA-B$^*$27:05 individuals, A2/M1$_{58}^+$CD8$^+$ T cells in HLA-B$^*$27:05-expressing donors are subdominant, with largely distinct TCR$\alpha\beta$ clonotypes and markedly reduced proliferative and polyfunctional capacity. Differences in TCR$\alpha\beta$ repertoires between immunodominant and subdominant A2/M1$_{58}^+$CD8$^+$ T cell populations might explain differential quantitative and qualitative characteristics of these influenza-responding CD8$^+$ T cells directed at the same A2/M1$_{58}^+$ epitope. Our data highlight the need to understand immunodominance hierarchies within individual donors across a spectrum of prominent virus-specific CD8$^+$ T cell specificities prior to designing T cell-directed vaccines and immunotherapies.

T cell-based vaccination strategies inducing cross-reactive CD8$^+$ T cell immunity have the potential to provide broad and long-lasting protection against distinct influenza A virus strains [7, 16, 17, 38], and even across all human influenza A, B and C viruses [15], including unpredictable, newly emerging strains with a pandemic potential. Ideally, the vaccine should encompass highly immunogenic epitopes for a diverse population with different HLAs. A number of prominent and "universal" (directed at viral peptides conserved over the last century) CD8$^+$ T cell specificities restricted by a range of HLA alleles have been identified [5, 14, 39–41], with A2/M1$_{58}$ being the most immunodominant, widely studied and highly prevalent in the global population. However, it is largely unknown how virus-specific CD8$^+$ T cell responses are elicited towards a vaccine cocktail of multiple universal influenza epitopes restricted by different HLAs, and whether one epitope induces immunodomination over others. Our study thus focused on understanding the immunodominance hierarchy for universal influenza-specific CD8$^+$ T cell responses to provide insights towards a rational design of broadly-protective universal influenza vaccines.

Immunodominance hierarchy patterns were dissected across universal HLAs (HLA-A$^*$02:01, -A$^*$03:01, -B$^*$08:01, -B$^*$18:01, -B$^*$27:05 and -B$^*$57:01), which provide broad population coverage worldwide [5]. Several studies have focused on HLA-A$^*$02:01-driven immune responses towards influenza virus infection [17, 32, 42, 43] directed towards the immunodominant A2/M1$_{58}$ epitope [41]. In context of other universal HLAs, A2/M1$_{58}^+$CD8$^+$ T cell responses remained the most immunodominant over A3/B8/B18 epitopes but were greatly reduced in individuals co-expressing HLA-B$^*$27:05 or HLA-B$^*$57:01. One study by Boon *et al.* showed that IFN-$\gamma^+$B27/NP$_{383}^+$ CD8$^+$ T cell responses were significantly higher than A2/M1$_{58}^+$ responses, although *in vitro* virus-expanded A2/M1$_{58}^+$ CD8$^+$ T cell responses were of the same magnitudes when compared to HLA-A2$^+$/non-B27 donors [44]. In our study, we found significant differences not only between B27/NP$_{383}^+$ CD8$^+$ and A2/M1$_{58}^+$ CD8$^+$ T cells in A2/B27 donors but also for A2/M1$_{58}^+$ CD8$^+$ T cells between A2/B27 and A2/non-B27 donors *ex vivo* and *in vitro*.

HLA-B$^*$27:05 has been associated with superior immune control during HIV [45, 46] and HCV-1 [47] infections. HLA-B$^*$27:05-restricted immune control was attributed to higher

levels of polyfunctionality, functional avidity and proliferation within HIV-specific B27-KK10$^+$CD8$^+$ T cells [48]. CD8$^+$ T cells displaying higher polyfunctionality improved vaccine efficacy towards vaccinia virus and Leishmania major [49–51]. High proliferative capacity of epitope-specific CD8$^+$ T cells was linked to better protection against HIV-1 infection [52]. CD8$^+$ T cells of high antigen sensitivity, thus increased functional avidity, could mediate effective viral clearance at low antigen concentrations [53–55]. Collectively, these studies provided key interdependent mechanisms for superior immune CD8$^+$ T cell responses. Our study demonstrated that immunodominant B27/NP$_{383}$$^+$CD8$^+$ T cells, in the context of influenza virus infection, were more superior to subdominant A2/M1$_{58}$$^+$CD8$^+$ T cells by exhibiting higher polyfunctionality, increased proliferative capacity and higher antigen sensitivity within HLA-A2$^+$/HLA-B27$^+$ co-expressed individuals. In contrast, A2/M1$_{58}$$^+$CD8$^+$ T cells in the absence of HLA-B27 were highly immunodominant displaying higher qualitative features that resemble the immunodominant B27/NP$_{383}$$^+$CD8$^+$ T cell response.

Given the qualitative and quantitative differences observed between immunodominant and subdominant A2/M1$_{58}$$^+$CD8$^+$ T cells, our *ex vivo* phenotypic analysis showed no differences in their T cell differentiation phenotype, which were mainly CCR7$^+$CD45RA$^-$ "central memory" T cells. Conversely, the immunodominant B27/NP$_{383}$$^+$CD8$^+$ T cell population had significantly higher proportions of CCR7$^-$CD45RA$^+$ "effector" cells compared to both A2/M1$_{58}$$^+$CD8$^+$ T cell populations, a phenotype that warrants further investigation as "effector" T cells were originally believed to be terminally differentiated and exhausted, but new evidence shows that effector T cells contribute to a pool of long-lived memory T cells [56, 57]. However, while HLA-A$^*$02:01$^+$M1$_{58}$$^+$ and HLA-B$^*$27:05$^+$NP$_{383}$$^+$ CD8$^+$ T cells are of different phenotypes, they do not differ in terms of CD107a expression, a marker of T cell degranulation and thus cytotoxicity.

We showed that the superior features of immunodominant A2/M1$_{58}$$^+$CD8$^+$ T cells compared to the subdominant CD8$^+$ T cell responses, in terms of response magnitude, polyfunctionality, avidity, proliferative capacity and precursor frequency, could be linked to the differences in their TCRαβ repertoires in both gene signatures and TCR diversity. Immunodominant A2/M1$_{58}$$^+$CD8$^+$ T cells predominantly consisted of the public TRAV27/TRBV19 TCR signature [17], which was absent or reduced in subdominant A2/M1$_{58}$$^+$CD8$^+$ T cells that displayed more diverse private repertoires and lower precursor frequencies. Similar features of subdominant A2/M1$_{58}$$^+$CD8$^+$ T cells from this study were recently described by our group in our aging elderly cohort showing a loss in prominent public TCRs but expansion of private suboptimal TCR clonotypes, which was associated with lower precursor frequencies [32, 58]. Similarly, we have found that immunodominant A68/NP$_{145}$$^+$ CD8$^+$ T cell responses were linked to highly expanded TCRαβ clonotypes, whereas subdominant responses in other individuals had more diverse non-expanded TCRαβ clonotypes [38]. Our findings are reminiscent of the HLA-B$^*$08:01/B$^*$44:02 trans-allele EBV model that impacted on the public LC13-TCR response against the HLA-B$^*$08:01-restricted FLRGRAYGL epitope, which was found in most HLA-B$^*$08:01 donors, but was explicitly replaced with a subdominant CF34-TCR response in HLA-B$^*$08:01$^+$/B$^*$44:02$^+$ donors [59, 60].

Our findings have implications for current and future T cell-based vaccine candidates and how the vaccine responses are assessed to further advance into clinical stages. Particularly in HLA-A2$^+$/B27$^+$ individuals, total CD8$^+$ T cell responses could be underestimated when measuring A2/M1$_{58}$$^+$CD8$^+$ T cell responses alone. This can be exemplified by a challenge study with the Modified Vaccinia virus Ankara (MVA) vaccine antigen consisting of complete NP and M1 from A/Panama/2007/99 joined by a 7 aa linker sequence (MVA+NP+M1) [61], where only A2/M1$_{58}$-tetramer CD8$^+$ T cell responses were measured [62]. Similarly, only M1-specific CD8$^+$ T cell responses, but not NP-specific responses, were assessed in tonsil MNCs following in vitro stimulation with the MVA-NP+M1 vaccine [63]. Furthermore, a T

cell-based vaccine candidate called Flu-v consists of 4 synthetic polypeptides, which includes the A2/M1$_{58}$ epitope but not the B27/NP$_{383}$ epitope [64]. Our study highlights the importance of considering the whole spectrum of immunodominant epitopes, other than A2/M1$_{58}$, in the design and assessment of vaccine-induced CD8$^+$ T cell responses.

Overall, in the presence of HLA-B27, we demonstrate a strong immunodominance hierarchy pattern of B27/NP$_{383}$$^+$CD8$^+$ T cell responses being far more superior than the A2/M1$_{58}$$^+$CD8$^+$ T cell response. Functional, quantitative and phenotypic characteristics within immunodominant and subdominant A2/M1$_{58}$$^+$CD8$^+$ T cells were underpinned by differential TCRαβ repertoires. These findings have implications in understanding and modulating antiviral and anticancer CD8$^+$ T cell responses generated by therapeutic immunizations or vaccinations.

## Methods

### Ethics statement

All experiments were conducted in accordance to the Australian National Health and Medical Research Council Code of Practice. Buffy coats (n = 30) were obtained from the Australian Red Cross Lifeblood. Peripheral bloods were obtained from healthy adults (n = 3) recruited at the University of Melbourne and Deepdene Medical Clinic, which were approved by the Human Research Ethics Committee (HREC) of the University of Melbourne (#1443389.4). All participants provided written informed consent prior to inclusion in the study. Human spleens from deceased organ donors (n = 10) were obtained following next-of-kin consent via DonateLife Victoria and approved by the Australian Red Cross Lifeblood Blood Service Ethics Committee (#2015#08). Cryopreserved PBMCs (n = 10) were obtained from Erasmus Medical Centre, Netherlands (approved by the Sanquin Bloodbank after informed consent was obtained).

### HLA typing and cell isolation

HLA-typing was performed by VTIS at Australian Red Cross Lifeblood (West Melbourne, VIC). PBMCs were freshly isolated by Ficoll-Paque density-gradient centrifugation and cryopreserved. Mononuclear cells (MNCs) were isolated from spleens as previously described [58]. The patient information is detailed in Table 2.

### Peptides, tetramers and APCs

Universal influenza peptides were purchased from GenScript (Hong Kong) Limited (Central, Hong Kong): A2/M1$_{58-66}$ (GILGFVFTL), A3/NP$_{265-273}$ (ILRGSVAHK), B8/NP$_{225-233}$ (ILKGKFQTA), B18/NP$_{219-226}$ (YERMCNIL), B27/NP$_{383-391}$ (SRYWAIRTR) and B57/NP$_{199-207}$ (RGINDRNFW). Peptide/MHC class I monomers A2/M1$_{58}$ and B27/NP$_{383}$ were generated in-house [59, 65] and conjugated at a 8:1 molar ratio with PE-streptavidin (SA) or APC-SA (BD Biosciences, San Jose, CA, USA) to form tetramers. Class I-reduced (C1R) cell lines were kindly provided by Prof. Weisan Chen (La Trobe University, VIC, Australia) (C1R-A$^*$02:01, C1R-B$^*$08:01), Prof. James McCluskey (University of Melbourne, VIC, Australia) (C1R-B$^*$57:01), Prof. Anthony Purcell (Monash University, Australia) (C1R-B$^*$27:05) and Dr. Nicole Mifsud (Monash University, VIC, Australia) (C1R-A$^*$03:01, C1R-B$^*$18:01).

### In vitro assays

Epitope-specific and virus-specific CD8$^+$ T cell lines were generated for 10–12 days in RF-10 media (+10 U/ml IL-2) using autologous peptide-pulsed or A/Puerto Rico/8/1934 virus-infected PBMCs as responder cells, as previously described [15]. T cell lines were restimulated

**Table 2. List of donors used in the study.**

| Donor code | Age | HLA-A | | HLA-B | | Source |
|---|---|---|---|---|---|---|
| D1 | 59 | 01:01 | 02:01 | 08:01 | 44:02 | Buffy Pack |
| D2 | ND | 01:01 | 02:01 | 08:01 | 35:01 | Buffy Pack |
| D3 | 53 | 01:01 | 02:01 | 08:01 | 27:05 | Buffy Pack |
| D4 | ND | 01:01 | 02:01 | 08:01 | 35:01 | Buffy Pack |
| D5 | 47 | 01:01 | 02:01 | 08:01 | 27:05 | Buffy Pack |
| D6 | 30 | 02:01 | 03:01 | 40:01 (60) | 56:01 | Buffy Pack |
| D7 | 25 | 02:01 | 03:01 | 07:02 | 40:01 | Buffy Pack |
| D8 | 54 | 02:01 | 03:01 | 35:01 | 57:01 | Buffy Pack |
| D9 | 50 | 02:01 | 68:01 | 27:05 | 15:18 | Buffy Pack |
| D11 | 64 | 02:01 | 03:01 | 27:05 | 13:02 | Buffy Pack |
| D12/SD1 | 50 | 02:01 | 26:01 | 07:02 | 27:05 | Buffy Pack |
| D13 | 37 | 02:01 | 03:01 | 07:02 | 40:01 | Buffy Pack |
| D14 | 42 | 02:01 | 03:02 | 18:01 | 35:08 | Whole Blood |
| D15/SD3 | 25 | 02:01 | 24:02 | 27:05 | 35:01 | Buffy Pack |
| D18 | 56 | 02:01 | | 07:02 | 57:01 | Buffy Pack |
| D19 | 58 | 02:01 | 24:02 | 27:05 | 35:01 | Spleen |
| D20 | 68 | 01:01 | 02:01 | 08:01 | 35:01 | Buffy Pack |
| D21/SD2 | 58 | 2 | 3 | 14 | 27 | Whole Blood |
| D22 | 49 | 03:01 | | 07:02 | 27:05 | Spleen |
| D23 | ND | 01:01 | 02:01 | 08:01 | 27:05 | Buffy Pack |
| D24 | 47 | 02:01 | 11:01 | 44:02 | 51:01 | Buffy Pack |
| D25 | 51 | 01:01 | 02:01 | 07:02 | 44:02 | Buffy Pack |
| D26 | 33 | 02:01 | 33:03 | 07:02 | 27:02 | Whole Blood |
| D27 | ND | 01:01 | 02:01 | 08:01 | 35:01 | Buffy Pack |
| D28 | ND | 01:01 | 02:01 | 08:01 | 35:01 | Buffy Pack |
| D29 | 45 | 02:01 | | 07:02 | 44:02 | Buffy Pack |
| D30 | ND | 01:01 | 02:01 | 08:01 | 27:05 | Buffy Pack |
| D31 | ND | 01:01 | 02:01 | 08:01 | 27:05 | Buffy Pack |
| D32 | 47 | 02:01 | 29:02 | 15:01 | 35:01 | Buffy Pack |
| D33 | 62 | 02:01 | | 07:02 | 44:02 | Buffy Pack |
| D34 | 67 | 01:01 | 02:01 | 13:02 | 44:32 | Buffy Pack |
| D35 | 22 | 02:01 | 11:01 | 15:02 | 54:01 | Buffy Pack |
| D36 | 69 | 02:01 | | 15:01 | 57:01 | Spleen |
| D37 | 57 | 02:01 | 11:01 | 15:01 | 35:01 | Spleen |
| D38 | 38 | 02:01 | 33:01 | 15:16 | 57:01 | Spleen |
| D39 | 33 | 02:01 | 03:01 | 35:01 | 51:01 | Spleen |
| D40 | 48 | 02:01 | 24:02 | 07:02 | 15:01 | Spleen |
| D41 | 65 | 02:01 | 29:02 | 27:05 | 44:03 | Spleen |
| D42 | 55 | 02:01 | 24:02 | 27:05 | 35:01 | Spleen |
| | | | | | | Lymph nodes |
| D43 | 65 | 02:01 | 24:02 | 27:05 | 35:01 | Spleen |
| | | | | | | Lymph nodes |
| D44 | 69 | 02:01 | 24:02 | 39:06 | 40:02 | Buffy Pack |
| D45 | 55 | 03:01 | 32:01 | 44:02 | | Buffy Pack |
| D46 | 22 | 02:01 | 68:01 | 44:02 | 51:01 | Buffy Pack |
| D47 | 29 | 01:01 | 24:02 | 08:01 | 14:02 | Buffy Pack |
| D48 | 31 | 02:01 | 03:01 | 14:02 | 58:01 | Buffy Pack |

(*Continued*)

**Table 2.** (Continued)

| Donor code | Age | HLA-A | | HLA-B | | Source |
|---|---|---|---|---|---|---|
| D49 | 30 | 03:01 | 24:02 | 35:03 | 44:02 | Buffy Pack |
| D50 | 41 | 03:01 | 11:01 | 3501 | 44:02 | Buffy Pack |
| D51 | 56 | 03:01 | 11:01 | 07:02 | 44:03 | Buffy Pack |
| D52 | 26 | 01:01 | 24:02 | 08:01 | 57:01 | Buffy Pack |
| D53 | 59 | 02:01 | 31:01 | 15:01 (62) | 44:02 | Buffy Pack |
| D54 | 35 | 01:01 | 03:01 | 35:01 | 57:01 | Buffy Pack |
| D55 | 69 | 02:01 | 32:01 | 07:02 | 44:03 | Buffy Pack |
| D56 | ND | 03:01 | | 07:02 | 15:01 | Buffy Pack |

ND, not determined. HLA subtypes in brackets cannot be ruled out.

with peptide-pulsed HLA-matched C1R cell lines during a 5–6 hour ICS assay, in the presence of GolgiPlug (BD Biosciences) [38]. For the degranulation assay, anti-CD107a AF488 (eBioscience #53-1079-42) and GolgiStop (BD Biosciences) were also added. Lymphocytes were then surfaced stained with anti-CD3-BV510 (Biolegend #317332) or anti-CD3 PE-CF594 (BD Biosciences #562280, anti-CD4 BV650 (BD Horizon #563875), anti-CD14 APC-H7 (BD Biosciences #560180), anti-CD19 APC-H7 (BD Biosciences #560177), anti-CD8 PerCPCy5.5 (BD Biosciences #565310) and Live/Dead Near-IR (Invitrogen), fixed, then intracellularly stained with anti-IFN-γ v450 (BD Horizon #560371) and anti-TNF APC (BD Biosciences #340534) using the Fixation/Permeabilization Solution Kit (BD Cytofix/Cytoperm, BD Biosciences). Samples were acquired on a BD Fortessa (BD Biosciences) and analyzed using Flowjo software version 10 (Treestar, OR, USA). Proliferation kinetics were measured on days d3, d6, d9 and d12 by ICS or tetramer staining to calculate the absolute numbers of epitope-specific cells. For all ICS staining, background staining was subtracted from the no peptide controls.

### Ex vivo assays

PBMCs and spleen MNCs from HLA-A*02:01+/B*27:05+ and HLA-A*02:01+/B*27:05- individuals were tetramer-stained for 1 hour at room temperature before undergoing dual tetramer-associated magnetic enrichment (TAME), as described [32], using the MACS PE- and APC-MicroBeads and LS columns (Milteny Biotec, Bergisch Galdbach Germany). Lymphocytes were surfaced stained with the above surface antibodies plus anti-CCR7 PECy7 (BD Pharmingen #557648) and anti-CD45RA FITC (BD Pharmingen #555488). Samples were acquired on a BD Fortessa or BD FACS Aria III for single-cell sorting and subsequent multiplex-nested RT-PCR for TCR analyses of paired CDR3α and CDR3β regions [32].

### Statistical analysis

Statistics were performed using GraphPad Prism software (San Diego, CA, USA) to compare between two (Mann-Whitney) or multiple groups (Kruskal-Wallis test, one-way ANOVA), as indicated in the figure legends where $p < 0.05$ was considered statistically significant. PBMCs and spleen MNCs were grouped together for statistical power (except Fig 2B).

### Supporting information

**S1 Data. Data source file for Figs 1–6.** Experimental raw data for Figs 1–6 are provided in S1 Data.
(XLSX)

## Acknowledgments

We greatly thank the DonateLife Victorian Coordinators for their assistance in obtaining spleens. We thank Profs. Weisan Chen, James McCluskey, Anthony Purcell and Dr. Nicole Mifsud for the C1R cell lines and ImmunoID Flow Cytometry Facility.

## Author Contributions

**Conceptualization:** Sneha Sant, Liyen Loh, Thi H. O. Nguyen, Katherine Kedzierska.

**Data curation:** Sneha Sant, Sergio M. Quiñones-Parra, Marios Koutsakos, Emma J. Grant.

**Formal analysis:** Sneha Sant, Thi H. O. Nguyen.

**Funding acquisition:** Katherine Kedzierska.

**Investigation:** Sneha Sant, Thi H. O. Nguyen, Katherine Kedzierska.

**Methodology:** Sneha Sant, Thi H. O. Nguyen, Katherine Kedzierska.

**Project administration:** Thi H. O. Nguyen, Katherine Kedzierska.

**Resources:** Thomas Loudovaris, Stuart I. Mannering, Jane Crowe, Carolien E. van de Sandt, Guus F. Rimmelzwaan, Jamie Rossjohn, Stephanie Gras.

**Software:** Sneha Sant, Thi H. O. Nguyen.

**Supervision:** Liyen Loh, Thi H. O. Nguyen, Katherine Kedzierska.

**Validation:** Sneha Sant, Thi H. O. Nguyen.

**Visualization:** Sneha Sant.

**Writing – original draft:** Sneha Sant, Liyen Loh, Thi H. O. Nguyen, Katherine Kedzierska.

**Writing – review & editing:** Sneha Sant, Thi H. O. Nguyen, Katherine Kedzierska.

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
