## [Decision Letter · Decision Letter 0]

26 May 2020

Dear A/Prof Kedzierska,

Thank you very much for submitting your manuscript "HLA-B*27:05 alters immunodominance hierarchy of universal influenza-specific CD8+ T cells" for consideration at PLOS Pathogens. As with all papers reviewed by the journal, your manuscript was reviewed by members of the editorial board and by several independent reviewers. The reviewers appreciated the attention to an important topic. Based on the reviews, we are likely to accept this manuscript for publication, providing that you modify the manuscript according to the review recommendations.

The reviewer's were enthusiastic about the manuscript and the significance of the studies to the field. They had minor comments for consideration.

Sincerely,

Stacey Schultz-Cherry

Guest Editor

PLOS Pathogens

Ana Fernandez-Sesma

Section Editor

PLOS Pathogens

Kasturi Haldar

Editor-in-Chief

PLOS Pathogens

orcid.org/0000-0001-5065-158X

Michael Malim

Editor-in-Chief

PLOS Pathogens

orcid.org/0000-0002-7699-2064

The reviewer's were enthusiastic about the manuscript and the significance of the studies to the field. They had minor comments for consideration.

Reviewer Comments (if any, and for reference):

Reviewer's Responses to Questions

**Part I - Summary**

Reviewer #1: In this work, the authors studied the immunodominance status of influenza-specific universal CD8+ T-cells in HLA-I heterozygote individuals, which express two or more universal HLAs for Influenza A. The article brings interesting data showing that, while CD8+ T-cell responses directed towards A2/M158 are generally immunodominant, this response is markedly diminished in individuals expressing also the HLA-B*27:05. The authors show that A2+B27- individuals present highly functional TRBV19/TRAV27 TCRab clonotypes, but that in B27+ persons, A2 cells present increased diversity and increased prevalence of private less functional clonotypes, explaining partially the differential immunodominance of responses. The immunodominance of B27 restricted T-cells has been already demonstrated over the B8 ones (Boon et al., 2002), and, in experimental mice models, the co-expression of B7 and B27 alleles is associated with a B7 immunodominant responses (Akram and Inman, 2013). Although the immunodominance in the context of influenza infection has been already demonstrated in several studies, this article is interesting in showing that HLA-B27 alters the immunodominance hierarchy and functional properties of universal influenza-specific CD8+ T cells, impacting the response of the most prevalent HLA across multiple ethnicities. The understanding of immunodominance in the context of Influenza infection is crucial for the development of T-cells based therapy. The manuscript is clear and well written. The present group has a recognize expertise and many high-quality publications in the field of CD8+ T cell responses in Influenza infection. I have only minor comments.

Reviewer #2: This is a very well-written manuscript investigating CD8 T cell immunodominance in HLA heterozygotes. The authors have found that A2/M1 CD8 T-cells in non-HLA-B*27:05 individuals were immunodominant, contained public TCR clonotypes and displayed highly polyfunctional and proliferative capacity. However, in HLA-B*27:05-expressing donors, A2/M1CD8 T-cells were subdominant, with distinct TCR clonotypes and diminished avidity, proliferation and polyfunctionality. This study will not only help us understand the protective CD8 T cell responses following influenza infection in different individuals, but also have significant impact on the future design of CD8 T cell-based influenza vaccines. The reviewer only has one minor comment regarding the manuscript: if the information is available to the authors, please list age, sex and ethnicity information of the blood donors in a table during revision.

**Part II – Major Issues: Key Experiments Required for Acceptance**

Reviewer #1: None

Reviewer #2: N/A

**Part III – Minor Issues: Editorial and Data Presentation Modifications**

Reviewer #1: • In keywords: correct the word “immunodominance”;

• In abstract, replace the full stop after “capacity” with a comma;

• In results, correct the text for Figure 2C (line 204). It’s indicating Figure 5C;

• About the data Figure 6 and related text , is it correct to conclude that B27 cells are more proliferative than A2 cells based on the numbers of tetramer+ cells and IFN-gamma production upon stimulation??

• Can the authors discuss the fact that A2 and B27 cells (i) present different phenotypes, (ii) althodo not differ in terms of cytotoxins expression?

Reviewer #2: If the information is available to the authors, please list age, sex and ethnicity information of the blood donors in a table during revision.

PLOS authors have the option to publish the peer review history of their article (what does this mean?). If published, this will include your full peer review and any attached files.

Reviewer #1: No

Reviewer #2: No
---

## [Editor Report · Decision Letter 1]

18 Jun 2020

Dear A/Prof Kedzierska,

We are pleased to inform you that your manuscript 'HLA-B*27:05 alters immunodominance hierarchy of universal influenza-specific CD8+ T cells' has been provisionally accepted for publication in PLOS Pathogens.

Best regards,

Stacey Schultz-Cherry

Guest Editor

PLOS Pathogens

Ana Fernandez-Sesma

Section Editor

PLOS Pathogens

Kasturi Haldar

Editor-in-Chief

PLOS Pathogens

orcid.org/0000-0001-5065-158X

Michael Malim

Editor-in-Chief

PLOS Pathogens

orcid.org/0000-0002-7699-2064
---

## [Editor Report · Acceptance letter]

28 Jul 2020

Dear A/Prof Kedzierska,

We are delighted to inform you that your manuscript, "HLA-B*27:05 alters immunodominance hierarchy of universal influenza-specific CD8^+^ T cells," has been formally accepted for publication in PLOS Pathogens.

Best regards,

Kasturi Haldar

Editor-in-Chief

PLOS Pathogens

orcid.org/0000-0001-5065-158X

Michael Malim

Editor-in-Chief

PLOS Pathogens

orcid.org/0000-0002-7699-2064